# The H^+^ Transporter SLC4A11: Roles in Metabolism, Oxidative Stress and Mitochondrial Uncoupling

**DOI:** 10.3390/cells11020197

**Published:** 2022-01-07

**Authors:** Joseph A. Bonanno, Raji Shyam, Moonjung Choi, Diego G. Ogando

**Affiliations:** Vision Science Program, School of Optometry, Indiana University, Bloomington, IN 47405, USA; rashyam@iu.edu (R.S.); choimoon@iu.edu (M.C.).; digogand@iu.edu (D.G.O.)

**Keywords:** glutamine, ammonia, corneal endothelial dystrophy, lactate, MCT4

## Abstract

Solute-linked cotransporter, SLC4A11, a member of the bicarbonate transporter family, is an electrogenic H^+^ transporter activated by NH_3_ and alkaline pH. Although SLC4A11 does not transport bicarbonate, it shares many properties with other members of the SLC4 family. SLC4A11 mutations can lead to corneal endothelial dystrophy and hearing deficits that are recapitulated in SLC4A11 knock-out mice. SLC4A11, at the inner mitochondrial membrane, facilitates glutamine catabolism and suppresses the production of mitochondrial superoxide by providing ammonia-sensitive H^+^ uncoupling that reduces glutamine-driven mitochondrial membrane potential hyperpolarization. Mitochondrial oxidative stress in SLC4A11 KO also triggers dysfunctional autophagy and lysosomes, as well as ER stress. SLC4A11 expression is induced by oxidative stress through the transcription factor NRF2, the master regulator of antioxidant genes. Outside of the corneal endothelium, SLC4A11’s function has been demonstrated in cochlear fibrocytes, salivary glands, and kidneys, but is largely unexplored overall. Increased SLC4A11 expression is a component of some “glutamine-addicted” cancers, and is possibly linked to cells and tissues that rely on glutamine catabolism.

## 1. Introduction

The membrane transporter, SLC4A11, is ubiquitously expressed in human and mouse tissues. Following the relatively recent discovery and cloning of the gene [1], several mutations have been linked to congenital hereditary endothelial dystrophy (CHED), a disorder of the corneal endothelium. Maintenance of corneal stromal hydration, required for corneal transparency and clear vision, is provided by the ion and fluid transport properties of the corneal endothelium. Therefore, interest in the normal function of SLC4A11 and the consequences of dysfunction within the corneal endothelium are of great interest. While there have been numerous studies on the physiological function of SLC4A11, this has focused mostly in the context of corneal endothelial cell dystrophy, where it appears to have the greatest deleterious effect when mutated in humans. In this review, we will often refer to corneal endothelial dysfunction related to SLC4A11, but also point out preliminary observations in auditory function, kidney, submandibular gland and in some carcinomas, in which there are emerging roles for SLC4A11.

## 2. SLC4A11 Gene and Protein Characteristics

Solute-linked cotransporter, SLC4A11, was first identified, cloned, and characterized by sequence homology with known members of the SLC4 family of membrane bicarbonate transporters [1]. Originally called BTR1 (bicarbonate transporter-related protein-1), the gene maps to chromosome 20p12, coding for 891 amino acids, and a protein of approximately 100 kDa that is increased by N-linked glycosylation [1,2,3]. SLC4A11 transcripts are strongly expressed in kidney, cornea, salivary gland, cochlea, trachea, thyroid, and testis [1,4,5]. Transcripts were also observed in ovary and lung squamous cell carcinoma [1] and recent reports suggest upregulation in ovarian carcinoma [6,7,8].

SLC4 bicarbonate transporters are classified as either Na^+^ coupled electroneutral, electrogenic HCO_3_^−^ cotransporters, or Na^+^-independent electroneutral Cl^−^/HCO_3_^−^ exchangers. While originally thought to be a bicarbonate transporter, SLC4A11 does not appear to transport bicarbonate, but rather is an electrogenic H^+^(OH^−^) permeation pathway [9,10,11,12]. All SLC4 members, including SLC4A11, share a similar membrane topology. SLC4A11 has a long hydrophilic N-terminal cytoplasmic domain, 14 transmembrane domains, extracellular loop glycosylation sites, and a short C-terminal hydrophilic cytoplasmic domain [1,2]. The glycosylated protein is detected at the plasma membrane while the core unglycosylated moiety is not. The glycosylated/unglycosylated SLC4A11 ratio is used as a marker for normal trafficking to the plasma membrane [13]. The long N-terminal cytoplasmic domain is characteristic of membrane proteins, including those in the SLC4 family. Mutations in the cytoplasmic domain can impair function. The removal of the cytoplasmic domain of SLC4A11 or replacement with GFP totally abrogates function. The replacement with the cytoplasmic domain of AE1 (SLC4A1), which has the highest homology with SLC4A11, partially rescues trafficking, but not function, indicating that the specific N-terminal sequence of SLC4A11 is essential [11,14]. SLC4A11 also appears to assemble as a dimer and binds stilbene derivatives similar to other SLC4 members, but the specific binding residues have not been determined [2,5,15,16,17]. SLC4A11 has a long third extracellular loop similar to the Na^+^-coupled SLC4 cotransporters, but has none of the characteristic cysteine residues [5]. SLC4A11 has the least sequence homology (20%) with other SLC4 members [5,12], which is consistent with its unique functionality as a H^+^ transporter. There is a similar level of sequence homology to BOR1, a boron transporter in plants [18,19]. This influenced initial physiological studies, suggesting that SLC4A11 had borate transport properties.

Three N-terminal variants of SLC4A11-A, -B, and -C have been identified with sequence differences only in exon 1, having 918, 891, and 875 amino acids, respectively [16,20]. The expression of the variants was studied in the corneal endothelium in order to understand which variant was prominently mutated that would lead to corneal endothelial dystrophy. Kao et al. [16] found that variant C was predominant in human corneal endothelium and that H^+^ permeation could be stimulated by the stilbene derivative DIDS. Variants B and C both located to the plasma membrane and had similar functional properties, whereas variant A was found to be intracellular [3,16]. In contrast, Malhotra et al. [20] found that SLC4A11-B (named v2) was most prominent in corneal endothelium. In either case, no functional differences between variants could be determined. Potentially, the N-terminal variants specify targeting SLC4A11 to specific plasma membrane domains or plasma membrane of intracellular organelles, but this remains unexplored.

## 3. Disease Associations

Several SLC4A11 homozygous or compound heterozygous mutations are associated with congenital hereditary endothelial dystrophy (CHED), a rare corneal endothelial dystrophy that can manifest in infancy [8,21,22,23,24,25,26]. CHED is characterized by progressive corneal edema, leading to degraded visual acuity. The only treatment is keratoplasty (corneal transplant). An examination of CHED specimens indicates progressive loss of endothelial cells with age, a thickening of the Descemet’s basement membrane of the endothelium, and evidence of increased oxidative stress within the remaining endothelial cells [27], a characteristic also observed in late-age Fuchs endothelial corneal dystrophy (FECD) [28,29]. CHED is sometimes associated with hearing deficits and is termed Harboyan syndrome, which appears to be due to a degeneration of cochlear fibrocytes [30,31,32,33]. Heterozygous SLC4A11 mutations have been associated with later-age FECD, which may be related to wild-type mutant heterodimers [13,17]. Parents of CHED children who are heterozygous for SLC4A11 mutations also appear to be at risk of developing FECD [34]. No other human disorders have been definitively associated with SLC4A11; however, there are some reports from animal models that suggest kidney involvement (polyuria) [32,35,36,37] and submandibular gland dysfunction [38], which is consistent with the strong expression of SLC4A11 in these tissues.

The characterization of CHED mutants has been studied in transfected HEK293 cells. The immunofluorescence evidence suggests that many mutants must be misfolded since they are retained in the endoplasmic reticulum [21]. FECD patient mutation analyses revealed heterozygous mutant and wild-type SLC4A11 dimers associated with the dystrophy. Comparing the abundance of mature glycosylated SLC4A11 with core unglycoslyated protein, and using NHS-biotin extracellular surface binding assays of HEK cells transfected with mutant SLC4A11, revealed significantly less mutant protein at the cell surface compared to WT [13,39]. Interestingly, CHED SLC4A11 mutants do not impair WT protein trafficking to the membrane; however, FECD-associated SLC4A11 mutants reduce WT protein membrane trafficking. This would explain the dominant inheritance of FECD [2,17]. Reducing the incubation temperature to 30 °C, permissive for some ER-retained proteins, allowed some mutants to move to the plasma membrane [2,40,41]. Additionally, diclofenac and other non-steroidal anti-inflammatory drugs have the potential to correct misfolding and allow the trafficking of mutant SLC4A11 to the plasma membrane [42]. These studies contrast with a more recent report, showing that five common SLC4A11 mutants are present in the plasma membrane at levels comparable to WT [43]. In this study, the mutants were transfected into PS120 fibroblasts and the membrane protein was referenced to the membrane protein Na^+^/K^+^-ATPase, rather than to GAPDH. These findings may be due to differences in expression systems and the approach used. An extensive analysis of mutants associated with either CHED or FECD was performed with the additional insight of a BRET (bioluminescence resonance energy transfer) assay for membrane localization [42]; it was suggested that CHED mutants are more likely to be ER-retained (60%) relative to FECD mutants (20%) in an HEK expression system. Interestingly, when transfected to polarized MDCK cells, HA-tagged SLC4A11 was in the lateral plasma membrane as expected, but most was found to be intracellular. Human CHED samples are very rare. However, a recent study examined SLC4A11 localization in two CHED endothelial specimens and found that SLC4A11 was localized to the plasma membrane and did not overlap with an ER marker [44]. Thus, it appears that trafficking and localization studies are greatly influenced by expression system, and that the use of corneal endothelial tissue and/or primary cultured endothelial cells, although more challenging, would give unequivocal answers.

## 4. SLC4A11 Knock-Out Models

Mouse gene knock-outs (KO) can provide valuable insight regarding the function of the gene product and secondary effects related to the loss of function of that gene. There have been three approaches to producing an SLC4A11 KO. The first used a retroviral gene trap vector, infecting embryonic stem cells, that integrated within *Slc4a11*. The resultant stem cell line was injected into C57BL/6J blastocysts [31]. The mice were outwardly normal. SLC4A11 was immunodetected in wild-type mice, but not KO. The loss of SLC4A11 in the cochlea led to vestibular labyrinth collapse and significantly abnormal auditory brain response and vestibular evoked potential. Although SLC4A11 was not detected in corneal cells of *Slc4a11^−/−^* mice, the corneal phenotype was mild. The endothelium was normal, there was no change in corneal thickness or appearance of edema; however, corneal basal epithelial cells were taller and comprised a greater percentage of total epithelial thickness. In contrast, using a β-galactosidase disrupted integration approach, Groger et al. [32] observed similar auditory changes and corneal epithelial cell changes but, in addition, there was a significant increase in corneal thickness, and the endothelial cell layer showed prominent intracellular vacuoles. Groger et al. found that SLC4A11 disruption within the kidney led to polyuria and hypo-osmotic urine. The third KO approach used a Cre-lox system, leading to the deletion of SLC4A11 exons 9 to 13 [15,35]. This approach yielded changes in auditory, renal, and corneal phenotypes. There were progressive increases in corneal thickness (see Figure 1), decreases in corneal endothelial cell density and endothelial cell vacuolation, increases in Descemet’s membrane thickness, and a reductions in endothelial cell regular hexagonal morphology. Subsequently, it was demonstrated that there was significant oxidative stress within *Slc4a11^−/−^* corneal endothelial cells [45,46]. Therefore, the KO phenotype recapitulates CHED and indicates that the loss of SLC4A11 function is sufficient to cause endothelial dystrophy. Interestingly, oxidative stress attributable to the loss of SLC4A11 causes lysosomal and autophagy dysfunction [47], which could also be a consequence of ER stress. As described below in “SLC4A11 is a Mitochondrial Uncoupler”, the major source of oxidative stress in the SLC4A11 KO is from mitochondrial dysfunction. In turn, this could produce ER stress and protein misfolding. If performed early, before there is significant cell loss and dysfunction, the mouse CHED model can be partially reversed by the delivery of the wild-type gene [48], or the systemic delivery of the antioxidant MitoQ that targets mitochondrial superoxide [47].

## 5. Gene Expression Changes in SLC4A11 Knock-Out

Conventional knock-outs provide a long term period during which gene expression could be altered from in utero to old age. SLC4A11 KO mice show corneal edema, stromal lactate accumulation, mitochondrial ROS, and alterations in extracellular matrix and junctional morphology at 7–12 weeks of age [35,45,46,49,50]. RNA-Seq analysis of freshly isolated corneal endothelium from 12-week-old WT and KO mice indicate several gene pathway expression alterations that shed light on a cause for the corneal edema [51]. Not surprisingly, given the increase in oxidative stress in the SLC4A11 KO, genes for glutathione metabolism were upregulated. Several genes for glycolytic enzymes and mitochondrial ETC components were downregulated along with the lactate transporter, MCT4. Atp1b2, a component of the Na/K-ATPase, was also downregulated. Using these findings, Ogando et al. [51] found reductions in lactate production, oxygen consumption, glutaminolysis, and Na/K-ATPase activity. There was also upregulation of several junctional proteins (e.g., occludins and claudins) but downregulation of Actin2, which can link cadherins to the actin cytoskeleton. Endothelium permeability to fluorescein diffusion was more than 50% greater in KO tissue, indicating a loss of the barrier function. Ogando et al. [51] concluded that the metabolic alterations, reduced lactate:H^+^ and Na/K-ATPase transport capacity, and alteration of the osmotic barrier function together can explain the significant edema seen in the SLC4A11 KO.

Using comparative transcriptome analysis of primary cultures of human corneal endothelial cells and SLC4A11 KO MCEC (mouse corneal endothelial cells), Zhang et al. [44] also found inhibition of cell metabolism, ion transport (e.g., SLC4A4-NBCe1), and mitochondrial function, along with the activation of AMPK-p53/ULK1, consistent with increased mitophagy and mitochondrial dysfunction. Downregulation of the sodium bicarbonate transporter (NBCe1) was a key finding with the cultured cells, but this was not seen in KO tissue [51]. Another key difference was that MCT4 (SLC16A4) was upregulated in cultured cells. These differences may be due to the culture environment versus in vivo tissue.

Gene knock-outs throughout embryonic development and early maturation provide ample time for many secondary effects that may be not be directly related to the gene of interest. To minimize the time for the evolution of secondary effects, determine the initial trigger, and understand the sequence of changes in this model, an inducible SLC4A11 KO was created [52]. Within a few weeks of induction, corneal edema morphological changes and oxidative stress were observed. Since the corneal endothelial ion and fluid transport mechanisms rely on a linkage with the efflux of lactate from the stroma [49], the stromal lactate concentration was determined. An examination of stromas from the conventional [50] and inducible [52] knock-outs revealed significant increases in stromal lactate concentration, as expected when the endothelium is dysfunctional. The initial trigger for pathology in the SLC4A11 KO is hypothesized to be mitochondrial ROS production. Future studies with the inducible model will help determine if this is true and elucidate the sequence of pathophysiological events that ultimately causes corneal edema and cell loss.

## 6. Ion Transport Function of SLC4A11

The cloning of SLC4A11 revealed some homology to the plant borate transporter AtBOR1. Potential borate transport by SLC4A11 was studied in transfected HEK cells. In the absence of borate, SLC4A11 was found to be an electrogenic Na^+^ and OH^−^(H^+^) transporter. In the presence of borate, Na^+^B(OH)_4_^−^ influx was observed [19]. The role of borate in mammalian cells is unclear; however, Park et al. suggested that it may have a role in cell growth and differentiation [18,19]. Using bovine corneal endothelial cells that express physiological levels of SLC4A11, Jalimarada et al. [10] found that SLC4A11 knockdown reduced Na^+^-dependent OH^−^(H^+^) permeability. However, borate had no effect on Na^+^-dependent effects on intracellular pH (pHi) or on intracellular [Na^+^] in bovine corneal endothelial cells [10]. Using SLC4A11 transfected HEK cells, Ogando et al. found that SLC4A11 did not transport HCO_3_^−^ or borate, but showed ethyl isopropyl-amiloride (EIPA)-sensitive Na^+^-OH^−^(H^+^) and NH_4_^+^ permeability [12]. EIPA inhibition of SLC4A11 H^+^ flux was also shown by Kao et al. [3]. As confirmed in two other studies, SLC4A11 does not transport borate [11,16]. In addition, there is an inconsistent demonstration of Na^+^-dependent and -independent electrogenic H^+^ flux by SLC4A11 in corneal endothelial cells [10] and in SLC4A11 transfected cell lines [3,10,12,16,53,54,55]. Consistent with the inward H^+^ flux from SLC4A11, mouse corneal endothelial cell lines from the SLC4A11 KO have a significantly higher pHi_i_ than WT cells when perfused in a low buffering capacity bicarbonate-free ringer. However, this difference is abolished when cells are perfused with a high buffering capacity bicarbonate ringer, indicating that the absolute inward H^+^ flux generated by plasma membrane SLC4A11 is relatively small [56].

The apparent enhancement of H^+^ flux when endothelial cells are perfused with NH_4_Cl [12] led to a series of patch-clamp studies to try to determine the nature of this enhanced flux. Using SLC4A11 transfected PS120 fibroblasts, Zhang et al. [53] demonstrated inward H^+^ currents in response to 10 mM NH_4_Cl that increased from pH 6.5 to 7.5 to 8.5. Further analysis indicated that the currents were increasing with increasing [NH_3_], but not [NH_4_^+^] or [H^+^]. If the [NH_3_] was held constant at each pH, the inward current was approximately the same, suggesting that SLC4A11 is activated by increasing [NH_3_] and not pH. These inward H^+^ currents were not Na^+^-dependent and were insensitive to EIPA [53]. Kao et al. [3] also showed NH_4_Cl-sensitive inward currents in SLC4A11-transfected HEK cells that were, in contrast, mildly inhibited by EIPA. Using two electrode voltage clamps of oocytes transfected with mouse Slc4a11, Loganathan et al. [11] observed similar inward H^+^ currents stimulated by NH_4_Cl, but concluded that SLC4A11 transports NH_3_ rather than NH_3_-H^+^. Myers et al. [54], also using transfected oocytes, showed enhanced Na^+^-independent H^+^ conduction, but concluded that SLC4A11 was activated by alkaline pH. Moreover, the H^+^ conductance of SLC4A11 was found to be steepest at pH 8.5 and it was found that mutants can shift the optimum pKa [55]. In summary, from these studies, it appears that SLC4A11 provides H^+^ conductance that can be either Na^+^ dependent or independent, is sensitive to NH_4_Cl, and is stimulated by alkaline pH. Zhang et al. [53] proposed that SLC4A11 is an NH_3_/H^+^ cotransporter. However, this does not rule out SLC4A11 being an NH_3_ activated H^+^ transporter, as demonstrating NH_3_ net fluxes is difficult. The mechanism of an NH_3_ activation is unknown, but could possibly be caused by changing the charge of pore amino acid residues that could also be accomplished by alkaline pH. A more recent study by Kao et al. [9], using SLC4A11 transfected HEK cells, demonstrated H^+^(OH^−^) conductive transport stimulated by alkaline pH. Ammonia-stimulated currents were also increased in alkaline pH. The shift in the reversal potential from NH_3_ suggested NH_3_-H^+^ cotransport was competing with H^+^(OH^−^) and the data, fitting a theoretical model of NH_3_-H^+^ and H^+^(OH^−^) interacting competitively within the transporter.

One of the features of the SLC4 family of transporters is their interaction with stilbene derivatives. SLC4A11 binding to stilbenes has been useful in membrane fractionation studies [40]. Whereas stilbenes generally inhibit transport activity, Kao et al. [16] showed an enhancement of H^+^ flux by the stilbenes, H_2_DIDS, SITS, and DIDS, which was also effective in increasing H^+^ transport of the R109H mutant. Zhang et al. [53] found that DIDS had no effect on NH_4_Cl-stimulated H^+^ currents. Interestingly, SLC4A11 appears to confer a modest amount of membrane water permeability [15]. Several mutants show diminished water flux [15,57]. However, in this case, water flux by wild-type SLC4A11 was shown to be inhibited by stilbenes [15]. Further pharmacological studies examining inhibition or stimulation of SLC4A11 in the context of the different transport modes of SLC4A11 are needed.

## 7. SLC4A11 Plasma Membrane Function

Given the properties of the SLC4A11 transporter, what might be the physiological function in corneal endothelial cells? In this regard, Parker [54] and Nehrke [58] have put forth a hypothesis, yet to be tested, that SLC4A11 facilitates lactate:H^+^ flux and the corneal endothelium pump by actively balancing pH_i_. SLC4A11 KO shows reduced lactate production and MCT4 expression in corneal endothelium, suggesting that there is some functional association [51,52] and that SLC4A11 activity is linked to lactate transport and specifically to MCT4, which is also on the basolateral membrane. H^+^ flux via SLC4A11 can proceed in either direction to balance pH_i_, according to Nehrke; however the electrochemical gradient favors H^+^ influx. A potential model (Figure 2) shows H^+^ influx via SLC4A11 facilitating lactate:H^+^ efflux into the lateral space via MCT4 with lactate diffusion across the leaky tight junction to the apical surface. Lactate efflux via MCT4 is in line with it being high affinity and capable of lactate export in high-lactate environments [59]. This lactate flux, combined with that from apical MCT2, could then provide the net lactate gradient between apical surface and basolateral space that drives water across the cells [49].

## 8. Facilitation of Glutamine Catabolism

The apparent stimulation of SLC4A11 activity by ammonia and high pH prompts the question of physiological relevance. Ammonia is produced by cells primarily as a by-product of amino acid metabolism, particularly glutamine catabolism. Direct production of ammonia occurs in glutaminolysis, the conversion of glutamine to glutamate catalyzed by glutaminase (GLS). Glutamate can subsequently be converted to αketoglutarate by glutamate dehydrogenase (GLUD), and release a second molecule of ammonia. This occurs primarily within mitochondria, where αketoglutarate can then enter the tricarboxylic acid (TCA) cycle and contribute to the production of reducing equivalents that drive the electron transport chain, increase oxygen consumption, and produce ATP. Increases in αketoglutarate can also drive the TCA cycle in reverse, a process called reductive carboxylation. Here, the goal is to produce citrate which feeds into biosynthetic processes needed for proliferating cells. This is often found in carcinomas that express the requisite increases in TCA cycle enzymes that facilitate reductive carboxylation [60]. Given the ammonia sensitivity of SLC4A11, a potential role in facilitating glutaminolysis was tested [45,56].

Using cultured human corneal endothelial cells and mouse corneal endothelium, which highly express SLC4A11, Zhang et al. found a high expression of plasma membrane transporters for glutamine and other amino acids, GLS1 and GLUD, and a second isoform of glutaminase, GLS2, in corneal endothelial cells [45]. Glucose and glutamine catabolism was then studied in the human corneal endothelial cells using stable-isotope-resolved metabolomics (SIRM), GC-MS. The TCA cycle intermediates were found to be derived from both glucose and glutamine in an approximate 50:50 ratio. Glutamine catabolism was verified by a significant production of ammonia in glutamine containing media versus glucose alone. Moreover, there was a significant increase in ATP production, suggesting that the TCA cycle is stimulated to produce more reducing equivalents. In addition, overall corneal endothelial fluid transport activity was greater in the presence of glutamine plus glucose versus glucose alone. When examining the SLC4A11 KO mouse corneal endothelium, Zhang et al. [45] found increased levels of Gls1, decreased Gls2, and significantly increased levels of nitrotyrosine staining, which is commonly used as a general marker of protein oxidation related to ammonia toxicity [61]. From the SLC4A11 wild-type and KO mice, Zhang et al. [56] then created mouse corneal endothelial cell (MCEC) lines. They found reduced proliferative capacity, increased expression of GLS1, and a significantly reduced amount of TCA cycle intermediates derived from glutamine, suggesting that SLC4A11 is facilitating glutamine catabolism. In summary, the data from these two studies indicate that SLC4A11 facilitates the use of glutamine in the TCA cycle and reduces ammonia-related oxidative stress. In turn, the studies would predict decreased oxygen consumption and ATP production in SLC4A11 KO, which point to a dysfunction of mitochondria.

## 9. SLC4A11 Is a Mitochondrial Uncoupler

To examine mitochondrial function, Ogando et al. [46] used SLC4A11-transfected PS120 fibroblasts, cultured human corneal endothelial cells, and SLC4A11 WT and KO mouse corneal endothelial cells (MCEC). SLC4A11 had been shown to have cytoplasmic localizations, as well as the plasma membrane [3,42]. Ogando et al. [46] found immunodetection of SLC4A11 from isolated mitochondria with an apparent inner mitochondrial membrane localization. When activated with ammonia, the mitochondrial membrane potential (MMP) depolarized, consistent with a flux of H^+^ into the inner mitochondrial matrix. WT and KO cells had similar oxygen consumption rates (OCR) when incubated in glucose alone. However, glutamine significantly increased OCR in WT, but not KO. OCR analysis indicated that glutamine induced a significant proton leak in WT that was absent in KO. Glutamine caused an increase in the NAD^+^/NADH ratio, indicating greater ETC activity, and increased ATP levels in WT, but not KO, consistent with the increased OCR. Mitochondrial reactive oxygen species (ROS) production was significantly greater in KO cells incubated in glutamine. In the absence of glutamine, mitochondrial ROS levels in KO cells were significantly lower than WT. The ROS levels paralleled the rate of apoptosis, which was highest in KO with glutamine, and lowest in WT and KO in glucose alone. The percentage of KO cells that were TMRE (tetramethylrhodamine) positive, indicating an intact MMP, dramatically decreased when incubated in glutamine, indicating ongoing mitochondrial damage. However, of the KO cells that were TMRE+ in glutamine, the staining intensity increased significantly with time, indicating MMP hyperpolarization. This suggested that glutamine catabolism in KO cells leads to increasing MMP hyperpolarization, causing increased ROS production and damage to mitochondria that then depolarizes. The damage appears to induce MTP opening (Figure 3), which would release pro-apoptotic factors. As such, KO cells in glutamine could be rescued with the mitochondrial-directed antioxidant, MitoQ; the chemical uncoupler, BAM15; or the Gls1 inhibitors, BPTES or CB839 [46]. Moreover, the addition of dimethyl-αketoglutarate, which feeds the TCA cycle directly, bypassing glutaminolysis and avoiding ammonia production, reduced KO cell apoptosis, decreased mitochondrial ROS, and increased ATP levels. Figure 4 shows a schematic of the model of SLC4A11 mitochondrial function. Glutamine catabolism greatly accelerates TCA cycle activity, producing reducing equivalents that in turn accelerate the ETC. A by-product of complex I and III activity is superoxide production that increases with hyperpolarizing MMP [62,63]. In addition, ammonia can have a direct catalytic effect on the complexes that increase superoxide production [46,64]. However, SLC4A11 activated by ammonia causes H^+^ influx, i.e., mitochondrial uncoupling, preventing MMP hyperpolarization and thereby reducing superoxide production.

## 10. Trafficking to Mitochondria

Proteins are directed to mitochondria either by an N-terminal pre-sequence signal peptide or by a chaperone-mediated pathway involving cryptic internal targeting sequences. Choi et al. [37], using in silico analysis and mass spectrometry of cross-linked proteins from PS120 transfected fibroblasts, determined that SLC4A11 does not have a mitochondrial pre-sequence, but utilizes a chaperone pathway. Both HSP90 and HSC70 participate in guiding SLC4A11 to the mitochondrial outer membrane complex, TOM70, which transports proteins into the intermembrane space, followed by inner membrane chaperones (TIMs) that guide the protein to the inner membrane. Inhibiting the HSP90 or HSC70 chaperone pathways in transfected cells and WT MCEC in turn inhibited the delivery of SLC4A11 to mitochondria, consistent with reduced glutamine dependent OCR, reduced ammonia production, and increased mitochondrial superoxide production. Moreover, Choi et al. [37] showed reduced OCR in mitochondria isolated from SLC4A11 KO MCEC and mouse kidney following inhibition of chaperone activity, confirming that the metabolic alterations seen in SLC4A11 deficient cells are not due to non-mitochondrial effects. Heat shock protein expression is typically increased by oxidative stress. Glutamine causes oxidative stress in corneal endothelial cells, so it was not surprising that Choi et al. [37] found that glutamine induced expression of HSP90, HSC70, and SLC4A11 in human corneal endothelial cells, suggesting that SLC4A11 is also an oxidative stress responsive protein.

## 11. SLC4A11 and Oxidative Stress

A hallmark of both FECD and CHED corneal endothelial dystrophies is evidence of cellular oxidative damage. Roy et al. [65] provided the first indication of an association of SLC4A11 with oxidative stress, showing that HEK cells transfected with mutant SLC4A11 had increased ROS formation in response to tert-butyl hydroperoxide. Expression of the mutants also demonstrated decreased mitochondrial activity, increased apoptosis, and reduced increases in antioxidant heme oxygenase, (HO-1), and NAD(P) H-quinoneoxidoreductase-1 (NQO1) expression. Concomitantly, there was a reduced nuclear factor erythroid-related factor-2 (NRF2) expression, the master regulator of these and other antioxidant genes. Guha et al. [27] showed that oxidative stress increased SLC4A11, NRF2, and HO-1 expression in human corneal endothelial cells, consistent with glutamine causing increased SLC4A11 expression [37]. Knockdown of WT SLC4A11 by siRNA increased ROS formation, decreased mitochondrial activity, and impaired increased expression of NRF2, HO-1, and NQO1 in response to oxidative stress. Post-surgical CHED specimens also showed reduced NRF2 and antioxidant gene expression [66]. These findings are consistent with reports of mitochondrial oxidative stress in the SLC4A11 KO mouse corneal endothelium [45,46,52]. Most recently, Guha et al. [66] showed conclusively that NRF2 activation increases SLC4A11 expression and that this is due to NRF2 binding to the SLC4A11 promoter, consistent with the notion that SLC4A11 is an oxidative stress protein.

SLC4A11 dysfunction, whether from mutants or a KO mouse, produces mitochondrial ROS that affects mitochondrial activity and can also affect neighboring cellular processes. Damaged mitochondria will prompt mitophagy, part of the general autophagy system. Shyam et al. [47] found that autophagy was activated in SLC4A11 KO MCEC as well as KO mouse endothelial tissue, relative to WT cells and fresh endothelial tissue. However, autophagy flux was aberrant due to reduced mass and poor functioning of lysosomes, the last step in the autophagy sequence where materials are degraded. The transcription factor, TFEB, that is needed to increase expression of lysosomes and other components of the autophagy chain, was significantly reduced. This could be reversed with the mitochondrially directed antioxidant MitoQ, indicating that the autophagy deficit was secondary to mitoROS. Moreover, systemic delivery of MitoQ could slow down the progression of corneal edema in young SLC4A11 KO mice, which is similar to the reductions in corneal edema seen when glutamine catabolism was circumvented with dimethyl αketoglutarate eye drops [46]. These results highlight the potential use of antioxidant therapeutic approaches for corneal endothelial dystrophies, for which there are clinical trials in place to treat FECD.

## 12. Studies of SLC4A11 in Non-Cornea Tissues

SLC4A11 has a ubiquitous expression profile, but besides cornea, there has not been much exploration of the functions. All of the SLC4A11 KO models found auditory deficits, leading to dysfunctional brain response and vestibular-evoked potential waveforms. Vestibular fibrocytes showed intracellular vacuolation [32] reminiscent of those seen in corneal endothelium [35,47]. Whether this is due to dysfunctional autophagy or lysosome production remains to be examined. SLC4A11 is expressed in the kidney medulla [32,36] and the knock-out yields urine hypo-osmolarity [32,35]. Based on anatomical location within the outer medulla, it is conceivable that SLC4A11 has a role in countercurrent ammonia transport [36]. There is high expression of SLC4A11 in mouse submandibular gland (SMG) diffusely localized in acinar and duct cells. The loss of SLC4A11 did not affect total saliva production, but yielded increases in [Na^+^] and [Cl^−^] in stimulated SMG saliva [38]. Interestingly, while not having a major role in pH_i_ regulation, the loss of SLC4A11 was linked to duct cell dropout and deficient NaCl reabsorption, especially in females. Whereas a model for the change in [Na^+^] and [Cl^−^] in stimulated SMG saliva could not be formulated, the diffuse localization of SLC4A11 in the SMG may suggest a mitochondrial-ER-autophagy defect that leads to the loss of duct cells.

## 13. SLC4A11 in Cancer

The Warburg effect, upregulation of aerobic glycolysis and production of lactic acid, is the most well-known metabolic property of cancer cells. In addition, many cancer cells require glutamine for proliferation, the so-called “Glutamine-addicted” carcinomas [67,68,69]. Glutamine supplies many substrates needed for growth including other amino acids, nucleic acid production, and ATP. An energized ETC also yields increased ROS production, which is thought to be mitigated by a combination of glutathione, NADPH production, and upregulation of antioxidant genes. SLC4A11 is highly expressed in HT29 and HCT116 colon carcinoma cell lines [46]. Transient siRNA knockdown of SLC4A11 (~45% of control expression in either line) increased mitoROS by 40% in both, while reducing ammonia production and cell proliferation [46]. Moreover, high expression of SLC4A11 appears to be a risk factor in ovarian cancer [6,7] and a recent study uncovered increased SLC4A11 expression associated with poor outcomes in colon cancer [70]. GLS1 inhibitors are currently in clinical trials for some cancers. These data suggest that inhibition of SLC4A11 activity, together with GLS1 inhibition, could provide a very robust disruption of glutamine-addicted cells.

## 14. Summary and Remaining Questions

SLC4A11 functions as an electrogenic H^+^(OH^−^) transporter. Whereas earlier studies suggested a Na^+^ dependence, most recent studies fail to show this [9,53,54]. The effect on intracellular pH regulation from a plasma membrane location appears to be small. Interestingly, there are two modes of activity for SLC4A11, the simple H^+^(OH^−^) conductance and the model of NH_3_-H^+^ and H^+^(OH^−^) interacting competitively within the transporter. SLC4A11 is present at the plasma membrane, mitochondria, and potentially other organellar membranes. The N-terminal variants may have a role in specifying targeting SLC4A11; however, this is not the case for mitochondrial targeting. Ammonia production within mitochondria activates inner mitochondrial membrane SLC4A11 H^+^ uncoupling. Is this due to NH_3_ (or alkaline?) activation of the pore or does NH_3_ diffuse from the matrix across the inner membrane and then participate in coupled NH_3_-H^+^ transport (Figure 4)? At the plasma membrane, the interaction of SLC4A11, in coordination with other ion fluxes, is unclear. In corneal endothelium, there appears to be a link with lactate:H^+^ cotransport. SLC4A11 KO causes reduced expression of MCT4, lactate production, and glycolytic activity in general. In WT cells, lactate production is increased in the presence of glutamine, which is significantly curtailed in the SLC4A11 KO [51]. This might be explained metabolically in that mitochondrial use of glutamine could divert pyruvate from mitochondrial entry to conversion to lactate. It is also tempting to speculate that NH_3_ produced from glutaminolysis activates plasma membrane SLC4A11, thereby facilitating lactate efflux, as envisaged in Figure 2. Lastly, only stilbene derivatives and EIPA have been tried as pharmacological modulators. EIPA has been consistently found to inhibit SLC4A11 activity. While stilbenes do bind to SLC4A11, they do not appear to possess inhibitory properties. Finding agonists and antagonists of SLC4A11 activity would benefit from a large-scale drug screening.

## Figures and Tables

**Figure 1 cells-11-00197-f001:**
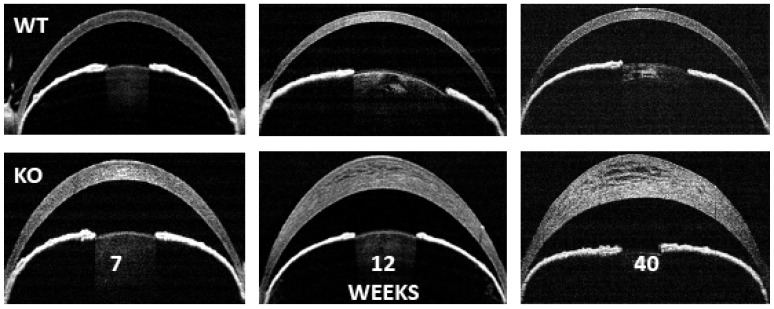
Corneal thickness of SLC4A11 wild-type vs. knock-out. Optical coherence tomography images of wild-type and SLC4A11 KO mice [35] at 7, 12, and 40 weeks, showing progressive corneal edema in KO.

**Figure 2 cells-11-00197-f002:**
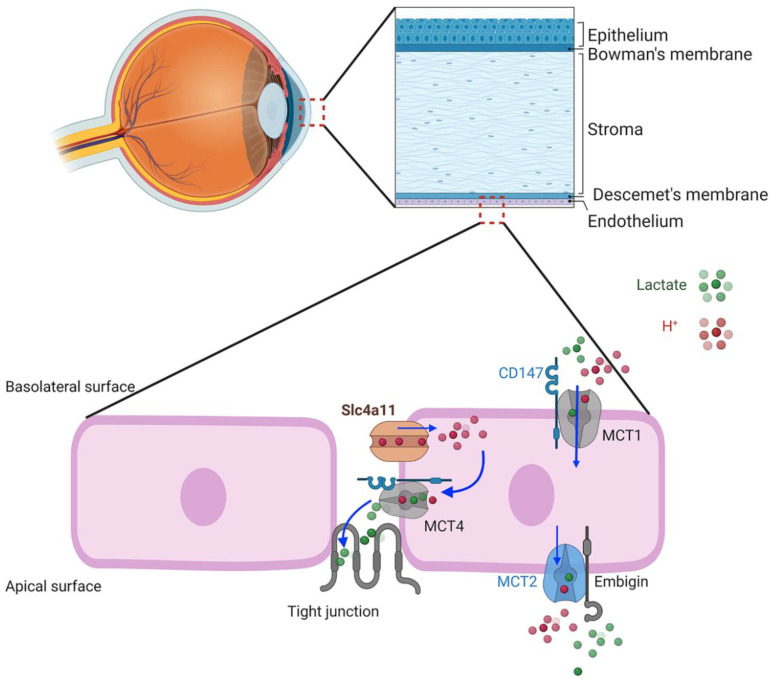
Schematic model of corneal endothelial lactate transport. MCT1 and MCT4 require the chaperone, CD147, that targets them to the basolateral membrane. The chaperone, embigin, targets MCT2 to the apical membrane. Basolateral SLC4A11 is associated with MCT4. H^+^ influx driven by the negative membrane potential feeds H^+^ to MCT4 lactate^−^:H^+^ cotransporter, loading paracellular space with lactate that diffuses across the leaky tight junction.

**Figure 3 cells-11-00197-f003:**
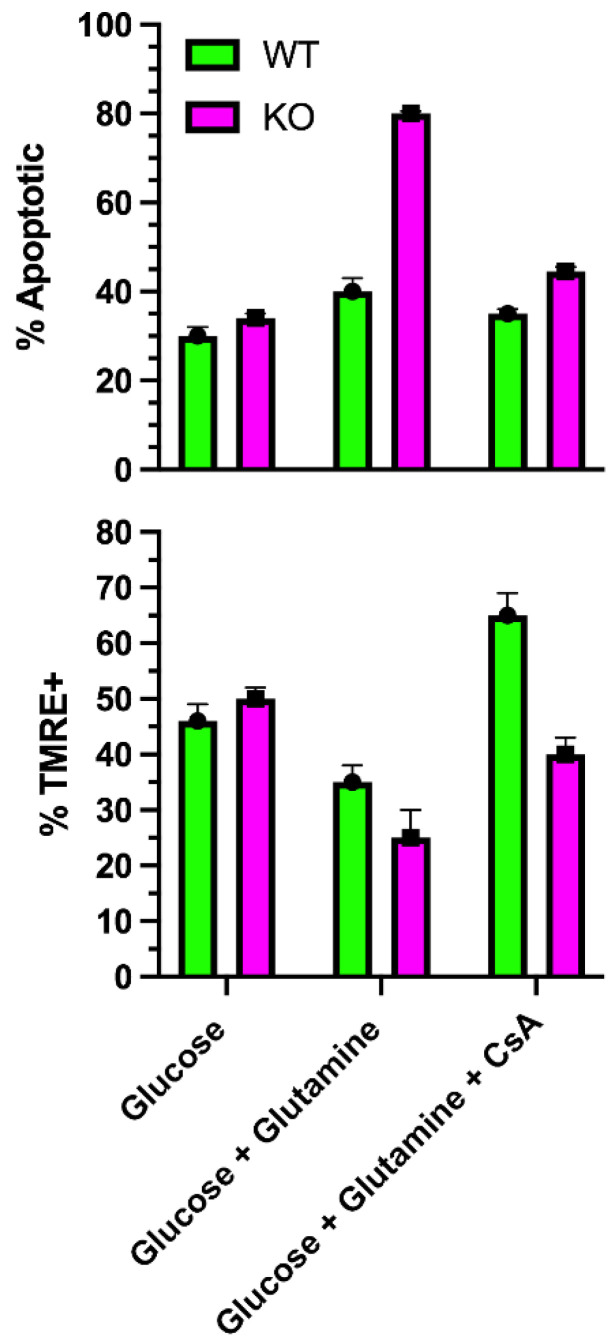
Effect of glutamine on mitochondrial transition pore opening. In SLC4A11 KO mouse corneal endothelial cells, glutamine causes a significant reduction in % TMRE+ cells, (i.e., more depolarized cells), and greater % apoptosis [46] (i.e., opening of MTP) that is reduced to WT levels by cyclosporine, which inhibits MTP opening (unpublished data).

**Figure 4 cells-11-00197-f004:**
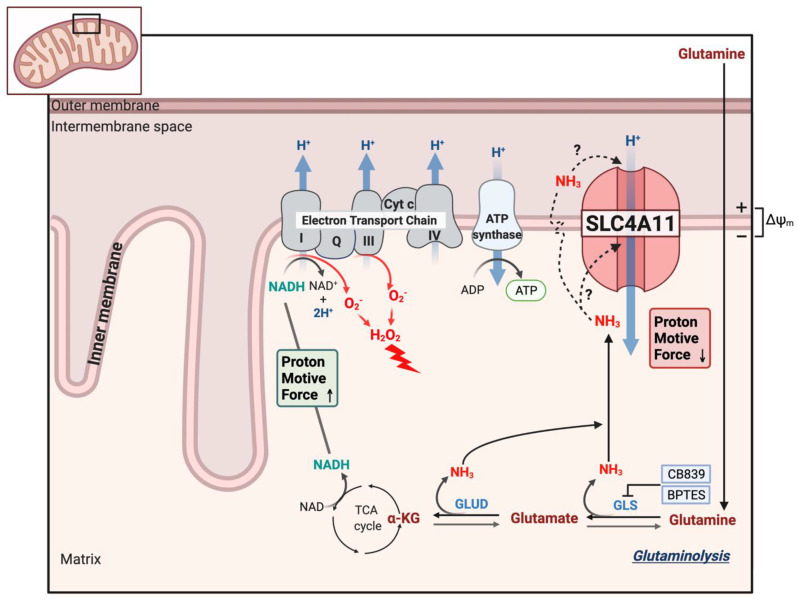
Model of SLC4A11 mitochondrial uncoupling. Glutaminolysis feeds the TCA cycle, producing reducing equivalents that accelerate the electron transport chain, increasing the formation of superoxide (O_2_^−^) radicals that can also be exacerbated by direct action of NH_3_ on complex I and III. Superoxide is quickly converted to the damaging hydrogen peroxide. Ammonia either directly activates SLC4A11 or diffuses across the inner membrane and cotransports with H^+^ into the matrix. The mild uncoupling action (H^+^ influx) of SLC4A11 prevents extreme MMP hyperpolarization, excess superoxide formation, mitochondrial damage/MTP opening, and apoptosis.

## Data Availability

Not applicable.

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
