# Peer review of "The H+ Transporter SLC4A11: Roles in Metabolism, Oxidative Stress and Mitochondrial Uncoupling"

_cells, 2022, doi:10.3390/cells11020197_

Round 1

Reviewer 1 Report

Since SLC4A11 was first reported as a causative gene of corneal endothelial CHED, it has been reported in many papers to have various functions. The authors' review, which covers the period since the SLC4A11 gene was reported to be involved in corneal endothelial diseases, includes unpublished data and may be the best review to date covering the latest reports on the SLC4A11 gene. In addition to corneal endothelium, the review also covers SLC4A11 kidney and cancer stem cell research, which is very interesting.

However, Vilas et al. (Vilas et al., Hum Mol Genet, 2013) reported that SLC4A11 is a water transporter and is involved in water transport in a cooperative manner with AQP1. Please comment on the reference.

SLC4A11 is thought to be expressed in mature corneal endothelial cells, but from what period in embryonic life is it expressed and is it involved in the homeostasis of corneal endothelium?

Author Response

Responses are shown in Italics.

Reviewer #1

Since SLC4A11 was first reported as a causative gene of corneal endothelial CHED, it has been reported in many papers to have various functions. The authors' review, which covers the period since the SLC4A11 gene was reported to be involved in corneal endothelial diseases, includes unpublished data and may be the best review to date covering the latest reports on the SLC4A11 gene. In addition to corneal endothelium, the review also covers SLC4A11 kidney and cancer stem cell research, which is very interesting.

However, Vilas et al. (Vilas et al., Hum Mol Genet, 2013) reported that SLC4A11 is a water transporter and is involved in water transport in a cooperative manner with AQP1. Please comment on the reference.

Yes, the potential SLC4A11 water permeability with this reference is discussed on page 6 in the context of stilbene interactions.

SLC4A11 is thought to be expressed in mature corneal endothelial cells, but from what period in embryonic life is it expressed and is it involved in the homeostasis of corneal endothelium?

This is a good question, but it has not been explored.

Reviewer 2 Report

Summary

 The manuscript of Bonnano et al. presents a comprehensive review of the SLC4A11. The manuscript focuses on corneal cells and the physiological data on SLC4A11. The manuscript describes proposed transport functions, presents historical views, which are then presented alongside recent data elucidating the ions transported and the uncoupling function of the protein. They conclude by placing this all in the context of the mitochondria and how this can impinge on mitochondrial and cellular metabolism. Lastly, ending the review with pathophysiological conditions that SLC4A11 mutation or absence may be involved in.

Brief Comments

The manuscript is well written, timely and well researched. For the most part, my comments are minor and only seek to improve the manuscript further.

Specific Comments

The authors can specifically improve some areas:

  • The authors should perhaps consider truncating the abstract to make it a more attractive and succinct summary of the review.
  • Line 68, the point of stilbene binding to SLC4A11 in the context of other SLC4A members, should be expanded to highlight if the binding region is conserved.
  • The point made at lines 80-83 should perhaps be presented more distinctly in the final section with the remaining questions section.
  • At lines 144-145, the auditory changes in Groger et al., were these the same as seen in the previous study? More description is needed there or a sentence highlighting more of the similarities and differences in this instance.
  • In line 158, there is an indication that mitochondrial dysfunction is the cause of oxidative stress, but the reader is not given any expanded information as to how this proposed mechanism works. This is explained later in the text, but here it seems out of place, and perhaps more explanation is needed here for clarity or some rewording.
  • In line 219, the authors state several studies but then only present two studies. They should be more explicit in their wording or include at least another reference.
  • At lines 221 and 222, perhaps consider incorporating corneal cells' references together with that statement, and transfected cells work with that statement. This would make it easier for the reader to follow, identify and refer to the works as appropriate.
  • In line 269, the authors state that SLC4A11 is functionally linked to MCT1 or MCT4, but that statement is not clear, and perhaps the authors can further expand. E.g. protein-protein interaction, messenger interaction or some other signalling mechanism. Especially since line 282 alludes to an interaction between SLC4A11 and MCT4
  • The sentence line 297-298 requires a reference.
  • The subtitle Cancer and SLC4A11 is not effective at present. It does not capture what is presented in this paragraph. A more appropriate subtitle might be SLC4A11 in Cancer Metabolism or something similar.

Minor points

  • Line 169 in utero should be in italics.
  • MCEC is not outlined as an abbreviation yet in the text. It is introduced later at line 315
  • Line 274- TJ should be specified as a tight junction

Figures & tables

  • Figure 2- this does a reasonably good job of summarizing the hypothesis, but its quality is blurry, making it difficult to appreciate some components of the figure.
  • Figure 4- the ATP synthase is not identified in the figure by name or otherwise. For clarity and readability, this should be corrected. Secondly, the production of superoxide anions from Complex I and Complex III is accurate; however, superoxide dismutates rapidly to hydrogen peroxide. Therefore, at the lightning bolt, it would be more accurate to indicate that hydrogen peroxide, as this species, also contributes to the oxidative stress mentioned in the text. Lastly, membrane potential should be delta psi to indicate the difference in potential across the membrane.

Author Response

Reviewer #2

Summary

 The manuscript of Bonnano et al. presents a comprehensive review of the SLC4A11. The manuscript focuses on corneal cells and the physiological data on SLC4A11. The manuscript describes proposed transport functions, presents historical views, which are then presented alongside recent data elucidating the ions transported and the uncoupling function of the protein. They conclude by placing this all in the context of the mitochondria and how this can impinge on mitochondrial and cellular metabolism. Lastly, ending the review with pathophysiological conditions that SLC4A11 mutation or absence may be involved in.

Brief Comments

The manuscript is well written, timely and well researched. For the most part, my comments are minor and only seek to improve the manuscript further.

Specific Comments

The authors can specifically improve some areas:

  • The authors should perhaps consider truncating the abstract to make it a more attractive and succinct summary of the review.

Abstract was shortened as suggested.

  • Line 68, the point of stilbene binding to SLC4A11 in the context of other SLC4A members, should be expanded to highlight if the binding region is conserved.

The specific residues have not been determined. This statement was added.

  • The point made at lines 80-83 should perhaps be presented more distinctly in the final section with the remaining questions section.

OK, added to final section.

  • At lines 144-145, the auditory changes in Groger et al., were these the same as seen in the previous study? More description is needed there or a sentence highlighting more of the similarities and differences in this instance.

Yes, they were similar. This is now stated.

  • In line 158, there is an indication that mitochondrial dysfunction is the cause of oxidative stress, but the reader is not given any expanded information as to how this proposed mechanism works. This is explained later in the text, but here it seems out of place, and perhaps more explanation is needed here for clarity or some rewording.

Thanks for pointing this out. We have added some wording to orient the reader.

  • In line 219, the authors state several studies but then only present two studies. They should be more explicit in their wording or include at least another reference.

This was fixed. Thanks.

  • At lines 221 and 222, perhaps consider incorporating corneal cells' references together with that statement, and transfected cells work with that statement. This would make it easier for the reader to follow, identify and refer to the works as appropriate.

Done.

  • In line 269, the authors state that SLC4A11 is functionally linked to MCT1 or MCT4, but that statement is not clear, and perhaps the authors can further expand. E.g. protein-protein interaction, messenger interaction or some other signalling mechanism. Especially since line 282 alludes to an interaction between SLC4A11 and MCT4

Thanks. This was reworded and clarified.

  • The sentence line 297-298 requires a reference.

Reference to reductive carboxylation added.

  • The subtitle Cancer and SLC4A11 is not effective at present. It does not capture what is presented in this paragraph. A more appropriate subtitle might be SLC4A11 in Cancer Metabolism or something similar.

Changed to SLC4A11 and Cancer

Minor points

  • Line 169 in utero should be in italics.
  • MCEC is not outlined as an abbreviation yet in the text. It is introduced later at line 315
  • Line 274- TJ should be specified as a tight junction

Fixed.

Figures & tables

  • Figure 2- this does a reasonably good job of summarizing the hypothesis, but its quality is blurry, making it difficult to appreciate some components of the figure.

This is a journal issue. The figure is high resolution

  • Figure 4- the ATP synthase is not identified in the figure by name or otherwise. For clarity and readability, this should be corrected. Secondly, the production of superoxide anions from Complex I and Complex III is accurate; however, superoxide dismutates rapidly to hydrogen peroxide. Therefore, at the lightning bolt, it would be more accurate to indicate that hydrogen peroxide, as this species, also contributes to the oxidative stress mentioned in the text. Lastly, membrane potential should be delta psi to indicate the difference in potential across the membrane.

Thanks. The figure was revised.